# A survey of patient and public perceptions and awareness of SARS-CoV-2-related risks among participants in India and South Africa

Oluchi Mbamalu[1]*, Surya Surendran[2¤a], Vrinda Nampoothiri[2¤b], Candice Bonaconsa[1], Fabia Edathadathil[2], Nina Zhu[3], Vanessa Carter[4], Helen Lambert[5], Carolyn Tarrant[6], Raheelah Ahmad[3,7], Adrian Brink[8], Ebrahim Steenkamp[9], Alison Holmes[3], Sanjeev Singh[2], Esmita Charani[1,3], Marc Mendelson[1]*

1 Division of Infectious Diseases & HIV Medicine, Department of Medicine, Groote Schuur Hospital, University of Cape Town, Cape Town, South Africa, 2 Department of Infection Control and Epidemiology, Amrita Institute of Medical Sciences, Amrita Vishwa Vidyapeetham, Kochi, Kerala, India, 3 Department of Medicine, National Institute for Health Research, Health Protection Research Unit in Healthcare Associated Infections and Antimicrobial Resistance, Imperial College London, London, United Kingdom, 4 Health Communication and Social Media, Johannesburg, South Africa, 5 Department of Population Health Sciences, Bristol Medical School, University of Bristol, Bristol, United Kingdom, 6 Department of Health Sciences, University of Leicester, Leicester, United Kingdom, 7 Division of Health Services Research and Management, School of Health Sciences, University of London, London, United Kingdom, 8 Division of Medical Microbiology, Faculty of Health Sciences, National Health Laboratory Service, Groote Schuur Hospital, University of Cape Town, Cape Town, South Africa, 9 Statistical Consulting Unit, Department of Statistical Sciences, University of Cape Town, Cape Town, South Africa

¤a Current address: Health Systems and Equity, The George Institute for Global Health, New Delhi, India
¤b Current address: Department of Health Sciences Research, Amrita Institute of Medical Sciences, Amrita Vishwa Vidyapeetham, Kochi, Kerala, India
* oluchi.mbamalu@uct.ac.za (OM); marc.mendelson@uct.ac.za (MM)

**Data Availability Statement:** Data underlying the study findings are included in the Supporting Information files submitted.

## Abstract

A cross-sectional survey among participants in India and South Africa to explore perceptions and awareness of SARS-CoV-2-related risks. Main outcome measures–proportion of participants aware of SARS-CoV-2, and their perception of infection risks as it related to their views and perceptions on vaccination, i.e., using COVID-19 vaccine uptake as proxy for awareness level. Self-administered questionnaires were used to collect data via web- and paper-based surveys over three months. Pearson's Chi-squared test assessed relationships between variables; a p-value less than 0.05 was considered significant. There were 844 respondents (India: n = 660, South Africa: n = 184; response rate 87.6%), with a 61.1% vs 38.3% female to male ratio. Post-high-school or university education was the lowest qualification reported by most respondents in India (77.3%) and South Africa (79.3%). Sources of pandemic information were usually media and journal publications (73.2%), social media (64.6%), family and friends (47.7%) and government websites (46.2%). Most respondents correctly identified infection prevention measures (such as physical distancing, mask use), with 90.0% reporting improved hand hygiene practices since the pandemic. Hesitancy or refusal to accept the SARS-CoV-2 vaccine was reported among 17.9% and 50.9% of respondents in India and South Africa, respectively; reasons cited included rushed vaccine development and the futility of vaccines for what respondents considered a self-limiting flu-like illness. In South Africa, vaccine acceptance was associated with improved hand

**Funding:** The work was supported by the Economic and Social Research Council (ESRC) as part of the Antimicrobial Cross Council Initiative supported by the seven UK research councils (Award Number: ES/P008313/1) (AH), the National Institute for Health Research – UK Department of Health in partnership with Public Health England (Award Number: HPRU-2012-10047) (AH) and the National Research Foundation of South Africa (Grant Number: 129755) (OM). The funders did not have any role in the study design and conduct, review or approval of the manuscript, or the decision to submit the manuscript for publication.

**Competing interests:** The authors have declared that no competing interests exist.

hygiene practices since the pandemic and flu vaccination in the preceding year. No relationship was noted between awareness and practice of infection prevention measures (such as hand hygiene) and socio-demographic factors such as employment status or availability of amenities. Pandemic response and infection prevention and control measures through vaccination campaigns should consider robust public engagement and contextually-fit communication strategies with multimodal, participatory online and offline initiatives to address public concerns, specifically towards vaccines developed for this pandemic and general vaccine hesitancy.

## Introduction

The SARS-CoV-2 pandemic has highlighted the importance of infection prevention at individual and community levels. The World Health Organization (WHO) has indicated that for public health infection prevention measures to be successful, all members of society (communities and professional groups included) should be fully engaged [1]. These measures include but are not limited to physical distancing, masking, hand hygiene, avoiding poorly ventilated indoor spaces, and isolation/quarantine if infected or exposed. For efficient buy-in and contribution to these measures, individuals should understand the risks, mode of viral transmission, and consequences of infection. As such, the success of infection prevention measures depends on individual and community-level awareness and the adoption of infection prevention behaviours, which in turn depends on their perceptions and cognizance of risk.

While effective public engagement has been highlighted as key to gaining buy-in for improved infection prevention practices [2–4], challenges related to socio-economic and other factors remain and may compromise infection prevention measures. This is especially so as the pandemic continues to evolve, with unequal effects and impact on different economies and populations [5, 6]. A compromise in infection prevention behaviour by one individual could cause reverberating consequences which can increase infection risks, not only for the individual but also for other people. Addressing infection prevention effectively requires due consideration of the prevailing context and associated insight into the factors that influence and by extension, can be used to motivate positive infection prevention behaviour. As such, additional research to explore public awareness, perceptions and behaviours about SARS-CoV-2 and how these may influence adherence to public health measures is needed, especially in low- and middle-income countries (LMIC).

The conceptual framework used to develop the survey was adopted from the Health Belief Model (HBM) [7]. According to the HBM, behaviours are adopted or changed by individuals when they perceive that the consequences of previous behaviours may be severe and/or they may benefit from the behaviour change. It also highlights the importance of barriers as a powerful predictor of behaviour change. Utilising such a framework contributes to better design of targeted questions, rather than development based on intuitive 'hunches' or 'guesses' [8]. We drew on this model in the development of a survey tool to be utilised across India and South Africa.

India (lower-middle-income) and South Africa (upper-middle-income) [9] are countries with emerging economies where the SARS-CoV-2 pandemic has had a significant impact [10]. Over time and in their respective continents, each of these countries had comparatively high burden of COVID-19 infections, as well as related complications and deaths [10]. Redeploying the capacity within an existing research collaboration focused on infection prevention and

control, and antibiotic stewardship across participating sites in these two countries [11, 12], we investigated the public's perceptions and awareness of SARS-CoV-2-related risks and infection prevention practices through analysis of data contributed by participants across the two countries.

## Methods

### Study design

We conducted a cross-sectional web- and paper-based survey. Data were collected by convenience sampling using a self-administered questionnaire. Any adult member (over 18 years old) of the public, who provided informed consent before participation, was eligible to participate. Respondents were resident in either of the participating countries (India or South Africa) at the time of participating in the survey. We had hypothesized that there would be no difference in the responses between participants in India and South Africa.

### Study development

A cross-sectional survey through online and offline tools was conducted from over a 3-month period. Report on the study was according to the STROBE guidelines [13] (S1 Checklist).

The research team–made up of pharmacists, physicians, nurses, social scientists, patient advocate and public engagement specialist, and quantitative data analysts–designed a 42-question survey to elicit information on the public's knowledge, perceptions and awareness of SARS-CoV-2 infection risks. The 4-part survey included participant demographics, general knowledge of SARS-CoV-2, perceived risks and barriers, and self-efficacy. In South Africa, survey questions and participant information leaflets were translated into IsiZulu, IsiXhosa and Afrikaans languages, whereas in India, the paper-based survey was translated into Malayalam for local distribution. The survey was piloted with members of the public, and relevant revisions were made before dissemination.

### Study settings and participant recruitment

The study was set in India and South Africa during the peak of the COVID-19 pandemic across both countries. The minimum sample size was calculated as 385 for each arm of the survey–India and South Africa. This will provide an estimate of the proportion of respondents who have knowledge, attitudes and perceptions of SARS-CoV-2 infection with 95% confidence and an alpha level of 0.05 to detect statistical significance, assuming that the expected proportion of respondents to have knowledge of COVID-19 is 50% (following unlimited population size application).

The survey was open for participation over a 3-month duration. Any member of the public (who was at least 18 years of age) resident in any of the two participating countries was eligible to participate. Voluntary response sampling, with some element of snowball sampling, was utilized to reach as many participants as possible. For online participation, individuals were informed of the survey through invitations (containing a link to the survey) sent by members of the research team to their various professional and personal networks. In addition, the researchers contacted representatives of various sectors of the public–who also helped to distribute the survey within their own networks. The offline/paper survey was conducted by researchers (assisted by medical social workers) who distributed copies of the questionnaire among participants (patients, patient carers and/or visitors) at the study site (hospital) in Kerala, India. Patients and patient carers who visited any specialty in the hospital at that time were invited (by researchers, assisted by medical social workers) to participate in the survey. All

invited individuals in the participating countries–both online and paper survey participants–received participant information leaflets, and those willing to participate had to provide informed consent before commencing the survey. Participation was voluntary across both countries.

In South Africa, the survey was available online in three languages–IsiXhosa, Afrikaans and English. In India, the survey was available online in the English language, and in the paper format in two languages, English and Malayalam–the prevalent local language at the study site in Kerala where the paper forms were distributed. During the survey development and dissemination period, use of the paper form for the survey was not possible in South Africa due to the COVID-19 restrictions at the time. In India, the paper forms were permissible for distribution following appropriate COVID-19 infection prevention measures; therefore, participants had a choice to participate either online or using the paper version of the survey.

## Data collection

Data from self-completed survey forms were collected by researchers from 15 September to 15 December 2020 and coincided with the first wave of the SARS-CoV-2 pandemic in India and the beginning of the second wave in South Africa. In both countries, the online version of the survey was available *via* the platform, Qualtrics.

In India, in addition to the online platform in English, paper survey questionnaires (in English and Malayalam) were also distributed among participants (patients, patient carers and/or visitors) at the study site (hospital) in Kerala–a 1350-bed tertiary academic hospital in an urban area [12]. While the hospital has different units for patients with different conditions, it was utilised mostly for COVID-19 cases during the pandemic. Patients and patient carers who visited any specialty in the hospital at that time were invited (by researchers, assisted by medical social workers) to participate in the survey. All invitations were issued while informing participants that participation was voluntary and that they could decline participation with no risk of prejudice. Participant selection at the hospital was not randomised; medical social workers who assist a lot in patient care at the study site went around the waiting areas of the hospital (the sections which were not under isolation due to COVID_19 measures) and invited participants to complete the survey. This was done as time allowed throughout the data collection period. Between 270 and 290 copies of the paper forms were shared to respondents. Some of the respondents returned more than one form–explaining that people around them had indicated interest and they had made additional copy/copies to share.

## Ethics statement

The study was approved by the relevant human research ethics committees at the Amrita Institute of Health Sciences, Kerala, India (Ref: IRB-AIMS-2020-232) and the University of Cape Town, South Africa (Ref: 311/2020). A patient advocate and public engagement specialist/civil society champion was involved in the design of the study material and also contributed as an author. Members of the public participated in the review of the survey tool and provided feedback for its modification. Formal consent was obtained prior to participation in the anonymous survey. For the online and paper versions of the survey, consent was indicated by the participant ticking the relevant box for consent on the survey form. Completion of the questionnaire (online or offline) after being provided with the Participant Information Leaflet (PIL) was also taken as an indication of informed consent to participate. All individuals were informed of their right to refuse to participate—and that there was no risk of prejudice attached to a refusal to participate.

### Statistical analyses

Data from participants who completed the paper-based format were captured in a Microsoft (MS) Excel file and codes assigned, while data of participants who completed the online form were exported to MS Excel. The data from the paper-based and online versions of the survey were cleaned and combined.

Descriptive statistics were used to report participant characteristics and survey responses. The underlying outcomes were awareness of the pandemic, perceived threats and barriers, and self-efficacy. Responses were captured as categorical variables, reported as percentages of received feedback for each item of interest (missing data were excluded) or, for certain questions, data were scaled from strongly agree to strongly disagree.

Pearson's Chi-squared test was used to assess relationships between variables and a logistic regression analysis was conducted with COVID-19 vaccination as the response variable; $p < 0.05$ was considered statistically significant for both tests. For logistic regression analysis, the variable was coded as **1** if people were willing to take the vaccine and **0** if otherwise. Both Pearson's Chi-squared and regression tests were conducted using R (version 3.6.2) [14].

## Results

### Participant demographics

There was a total of 844 respondents (660 participants from India and 184 participants from South Africa)–S1 Data). There were 318 respondents to the online survey and 342 patients or patient carer respondents to the paper survey in India. The response rate for the online survey was 87.6% (502/573), calculated as the ratio of participants who clicked on the survey link versus those who commenced participation. The response rate for the paper version of the survey could not be estimated, as respondents returned a higher number of the completed survey forms than the initial number disseminated, indicating the forms had been copied and shared more widely.

There were more female (515/844, 61.0%) than male (323/844, 38.3%) respondents (Table 1). Three entries for age were excluded (one was invalid with two selections and two were missing), resulting in a total response of 657 for age entries. Most of the respondents in India and South Africa were in the 20-29-year (310; [n = 657] 47.2%) and 40-49-year (57; 31.0%) age groups, respectively.

Majority of the participants across both countries had post-high school education (46.7%) (Table 1). Five participants selected **Other** in response to the question **What is your highest educational qualification?**, and noted **Other** educational qualifications as related to a Diploma in Elementary Education, Secretarial diploma, previous radiography teaching experience, Supplier Relationship Management (SRN)/Supply Chain Management (SCM), and one had no further information.

The percentage of student respondents was higher in India (21.3%, 135/633) than South Africa (3.0%, 5/166). Unemployment was higher among respondents in India (19.3%, 122/163) than in South Africa (7.8%, 13/166), while there were more self-employed (30.1%, 50/166) and retired (11.4%, 19/166) respondents in South Africa.

### Knowledge and concerns of SARS-CoV-2 transmission and infection

Reported sources of SARS-CoV-2 information, completed by 652 and 172 participants in India and South Africa, respectively, are shown in Fig 1A and 1B). Across both countries, traditional news channels and media and journal publications (India, 75.6%; South Africa, 64.0%) were the most common sources of pandemic-related information among respondents,

**Table 1. Self-reported respondent demographics.**

| Characteristic | India | South Africa | Total |
|---|---|---|---|
| | n (%) | n (%) | n (%) |
| **Country of residence** | 660 (78.2) | 184 (21.8) | 844 (100) |
| **Province / State or territory of residence** | Kerala: 400 (60.6) | WC: 92 (50.0) | 492 (58.3) |
| | Others: 256 (38.8) | Others: 84 (45.7) | 340 (40.3) |
| | Missing: 4 (0.6) | Missing: 8 (4.3) | 12 (1.4) |
| **Gender** | **(n = 660)** | **(n = 184)** | **(n = 844)** |
| Male | 285 (43.2) | 38 (20.7) | 323 (38.3) |
| Female | 369 (55.9) | 146 (79.3) | 515 (61.1) |
| Prefer not to say | 5 (0.8) | 0 | 5 (0.6) |
| Missing | 1 (0.2) | 0 | 0 |
| **Age** | **n = 657 (%)** | **n = 184 (%)** | **n = 841 (%)** |
| Younger than 20 years | 41 (6.2) | 7 (3.8) | 48 (5.7) |
| 20 to 29 years | 657 (47.2) | 14 (7.6) | 324 (38.5) |
| 30 to 39 years | 133 (20.2) | 21 (11.4) | 154 (18.3) |
| 40 to 49 years | 82 (12.5) | 57 (31.0) | 139 (16.5) |
| 50 to 59 years | 49 (7.5) | 49 (26.6) | 98 (11.7) |
| 60 to 69 years | 30 (4.6) | 27 (14.7) | 57 (6.8) |
| 70 years and older | 12 (1.8) | 9 (4.9) | 21 (2.5) |
| **Regular water supply** | **n = 650 (%)** | **n = 176 (%)** | **n = 826 (%)** |
| Yes | 560 (86.2) | 172 (97.7) | 732 (88.6) |
| No | 90 (13.8) | 4 (2.3) | 94 (11.4) |
| **Education** | **n = 653 (%)** | **n = 175 (%)** | **n = 828 (%)** |
| Primary schooling | 28 (4.3) | 1 (0.5) | 29 (3.5) |
| Secondary schooling | 113 (17.3) | 25 (13.6) | 138 (16.7) |
| Post-high school | 291 (44.6) | 96 (52.2) | 387 (46.7) |
| Post-graduate degree | 219 (33.5) | 50 (27.2) | 269 (32.5) |
| Other | 2 (0.3) | 3 (1.6) | 5 (0.6) |
| **Employment** | **n = 633 (%)** | **n = 166 (%)** | **n = 799 (%)** |
| Student | 135 (21.3) | 5 (3.0) | 140 (17.5) |
| Employed, part time | 35 (5.5) | 11 (6.6) | 46 (5.8) |
| Employed, full time | 249 (39.3) | 62 (37.3) | 311 (38.9) |
| Self-employed | 47 (7.4) | 50 (30.1) | 97 (12.1) |
| Retired/Pensioner | 27 (4.3) | 19 (11.4) | 46 (5.8) |
| I was furloughed/laid off during the lockdown | 8 (1.3) | 5 (3.0) | 13 (1.6) |
| Unemployed | 122 (19.3) | 13 (7.8) | 135 (16.9) |
| Other | 10 (1.6) | 1 (0.6) | 11 (1.4) |

* 400 (from 58 and 342 respondents to the online and paper versions of the survey)

WC[#]: Western Cape province of South Africa

along with social media (India, 65.6%; South Africa, 60.5%), government websites (India, 44.8%; South Africa, 51.7%), and family and friends (India, 48.5%; South Africa, 44.8%). Among those who indicated use of social media for pandemic-related information, Facebook (India, 77.3%; South Africa, 82.7%) and WhatsApp (India, 84.6%; South Africa, 52.9%) were the most frequently used sites as shown in Fig 1A and 1B.

In Table 2, the respondents' knowledge of SARS-CoV-2 transmission routes, infection course and prevention/management options is summarised. The primary route of SARS-CoV-2 transmission identified was nasal/oral droplets, airborne particles, and infected body fluids.

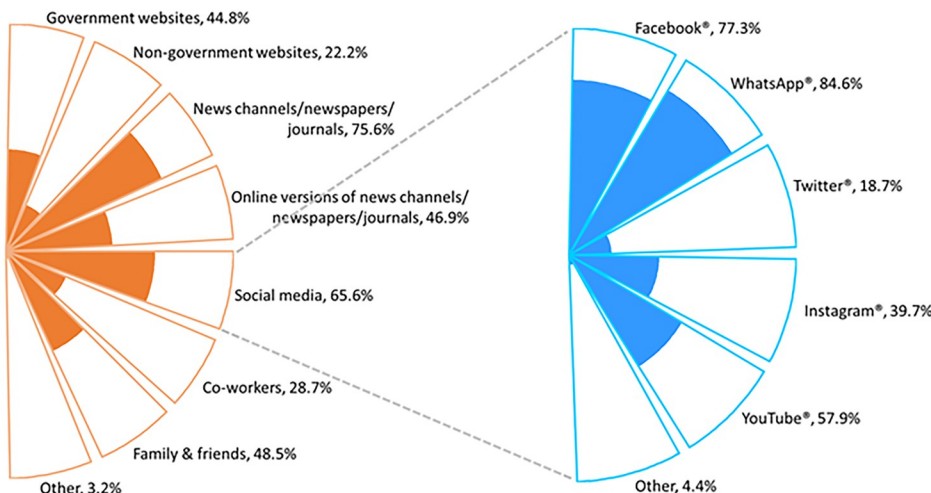

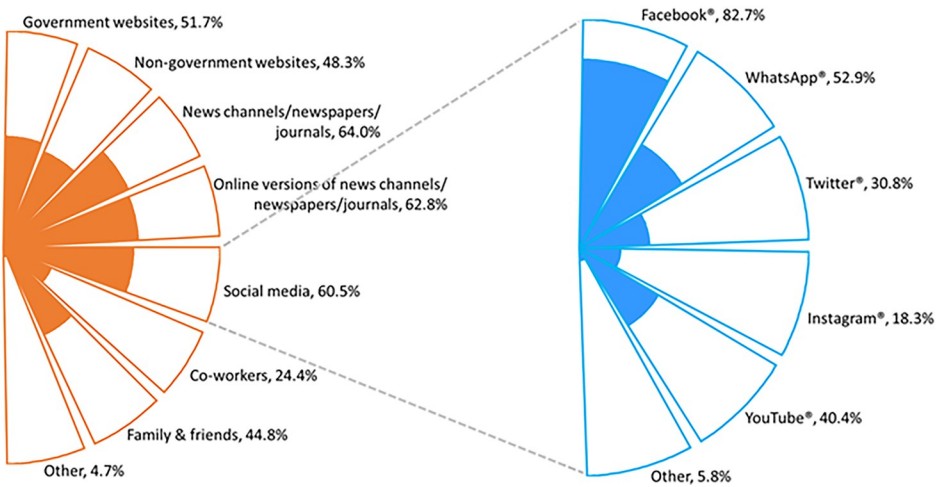

**Fig 1. Respondents' sources of SARS-CoV-2 information.** From A: general (n = 652) and social media in India; B: general (n = 172) and social media in South Africa.

More than half of the respondents also demonstrated knowledge of SARS-CoV-2 incubation and symptom manifestation, quarantine objectives, and general duration of isolation for infected patients.

More frequent hand washing was reported across both countries (90.0%); however, a higher percentage of respondents in South Africa (13.8%) than in India (2.9%) noted no difference in their hand hygiene practices. Overall, 75.0% of all the respondents indicated their willingness

**Table 2. Respondent's knowledge and experiences of the pandemic.**

| | Response (%) | | |
|---|---|---|---|
| **Knowledge of SARS-CoV-2** | **India** | **South Africa** | **Total** |
| **Major routes of transmission** | | | |
| Infected bodily fluids | 426/516 (82.6) | 55/119 (46.2) | 481/635 (75.7) |
| Nasal or oral droplets | 555/591 (93.9) | 158/165 (95.8) | 713/756 (94.3) |
| Airborne | 352/458 (76.9) | 114/147 (77.6) | 466/605 (77.0) |
| Foodborne | 113/328 (34.5) | 9/101 (8.9) | 122/429 (28.4) |
| Waterborne | 119/322 (37.0) | 5/98 (5.1) | 124/420 (29.5) |
| Other (please specify) | 11/60 (18.3) | 5/ 25 (20.0) | 16/85 (18.8) |
| **Time to symptom onset** | n = 626 (%) | n = 171 (%) | n = 797 (%) |
| Immediately–there is no delay | 42 (6.7) | 1 (0.6) | 43 (5.4) |
| 0 to 2 weeks | 448 (71.6) | 152 (88.9) | 600 (75.3) |
| 2 to 4 weeks | 80 (12.8) | 12 (7.0) | 92 (11.5) |
| Over 4 weeks | 10 (1.6) | 2 (1.2) | 12 (1.5) |
| I don't know | 35 (5.6) | 4 (2.3) | 39 (4.9) |
| Multiple entries | 11 (1.8) | 0 | 11 (1.4) |
| **Perceived reason for quarantine of SARS-CoV-2-positive individuals** | n = 625 (%) | n = 171 (%) | n = 796 (%) |
| To help them get better | 22 (3.5) | 0 | 22 (2.8) |
| To prevent them from infecting others | 430 (68.8) | 124 (72.5) | 554 (69.6) |
| There is no good reason for that | 3 (0.5) | 12 (7.0) | 15 (1.9) |
| Other, please specify | 1 (0.2) | 2 (1.2) | 3 (0.4) |
| I don't know | 9 (1.4) | 0 | 9 (1.1) |
| Multiple entries | 159 (25.4) | 33 (19.3) | 192 (24.1) |
| **Duration of isolation (if not admitted to a healthcare facility)** | n = 625 (%) | n = 171 (%) | n = 796 (%) |
| As soon as coughing stops | 5 (0.8) | 0 | 5 (0.6) |
| 10 to 14 days after symptoms first started | 314 (50.0) | 136 (79.5) | 450 (56.5) |
| As soon as they feel better | 36 (5.7) | 3 (1.8) | 39 (4.9) |
| 21 days after symptoms stop | 110 (17.5) | 9 (5.3) | 119 (14.9) |
| For asymptomatic cases: as advised by healthcare guidelines | 61 (9.7) | 10 (5.8) | 71 (8.9) |
| They do not need to be isolated | 3 (0.5) | 9 (5.3) | 12 (1.5) |
| I don't know | 28 (4.5) | 4 (2.3) | 32 (4.0) |
| Multiple entries | 71 (11.3) | 0 | 71 (8.9) |
| **Changes in hand washing practices** | n = 615 (%) | n = 167 (%) | n = 782 (%) |
| I wash/sanitise my hands more often | 561 (91.2) | 143 (85.6) | 704 (90.0) |
| I wash/sanitise my hands less often | 13 (2.1) | 0 | 13 (1.7) |
| There is no difference in how often I wash/sanitise my hands | 18 (2.9) | 23 (13.8) | 41 (5.2) |
| Other, please specify | 2 (0.3) | 1 (0.6) | 3 (0.4) |
| I don't know | 6 (1.0) | 0 | 6 (0.8) |
| Multiple entries | 15 (2.4) | 0 | 15 (1.9) |
| **Avoided visit to healthcare facility because of SARS-CoV-2** | n = 594 (%) | n = 161 (%) | n = 755 (%) |
| Yes | 171 (28.8) | 60 (37.3) | 231 (30.6) |
| No | 353 (59.4) | 70 (43.5) | 423 (56.0) |
| Not applicable/had no need to visit a healthcare facility | 70 (11.8) | 31 (19.3) | 101 (13.4) |
| **Would you have a SARS-CoV-2 vaccination?** | N = 587 (%) | n = 161 (%) | n = 748 (%) |

(*Continued*)

**Table 2.** (Continued)

| | Response (%) | | |
|---|---|---|---|
| **Knowledge of SARS-CoV-2** | **India** | **South Africa** | **Total** |
| Yes | 482 (82.1) | 79 (49.1) | 561 (75.0) |
| No | 36 (6.1) | 55 (34.2) | 91 (12.2) |
| I don't know | 69 (11.8) | 27 (16.8) | 96 (12.8) |

to receive vaccination when it becomes available; however, the percentages were higher in India (82.1%) than in South Africa (49.1%). The most common reasons cited for not accepting vaccination were perceptions of rushed vaccine development and the futility of vaccines for what respondents considered a self-limiting flu-like illness.

## Self-efficacy: Perceptions of SARS-CoV-2 infection prevention measures

Respondents' perceptions and concerns about their ability to cope with SARS-CoV-2 infection prevention measures are presented in Fig 2, given their perceived knowledge and awareness of the pandemic and infection risks. More than half of respondents in each country reported that they have sufficient knowledge of SARS-CoV-2, understood available information on the pandemic, would know what to do or questions to ask if they or someone else contracted SARS-CoV-2, have access to healthcare were they to become ill with SARS-CoV-2 infection, and would be able to cope with extended containment measures such as a lockdown. Compared to South Africa, more respondents in India reported concern over infection, its financial implications and associated stigma. On the intent to wear a face mask, 8.6% and 26.2% of respondents in India and South Africa reported dissatisfaction with this measure while outdoors, respectively.

There was no association between hand washing and water supply (Table 3), as even those without access to water supply reported that they washed their hands more frequently since the pandemic ($p = 0.2168$ and $p = 0.7970$ in India and South Africa, respectively). Water supply showed a mixed relationship with employment as some full-time workers had no access to water. The test highlights a difference between participants in the two countries; $p = 0.0008$ and 0.4471 for India and South Africa, respectively.

Among survey respondents in South Africa, we noted a significant difference in vaccine acceptance/hesitancy between those who indicated that they practice HH more often since the pandemic compared to those who indicated that there has been no difference in their HH practices since the pandemic ($p = 0.05$) (S2 and S3 Data). There was also a significant difference in vaccine acceptance/hesitancy between those who were and were not vaccinated against the influenza virus ($p = 0.003$). Across both countries, there was a significant difference ($p < 0.001$) in vaccine acceptance/hesitancy between self-employed individuals and students.

Our results show that some respondents avoided healthcare facilities during this pandemic (S2 and S3 Data). Some participants in this study reported avoiding healthcare facilities because of a fear of contracting the COVID-19 virus; employment status did not seem to have a significant relationship with this fear although more of a case could be made for South Africa ($p = 0.1686$) as it had a lower p-value than India ($p = 0.3143$)–S2 Data. While the p-value was still too low to be considered significant, there were less South African participants than Indian participants, therefore with more data this relationship may turn out to be significant.

## Discussion

This study provides insight into the public's awareness and perspectives of the SARS-CoV-2 infection, risks and preventive practices in two middle-income countries hard hit by the

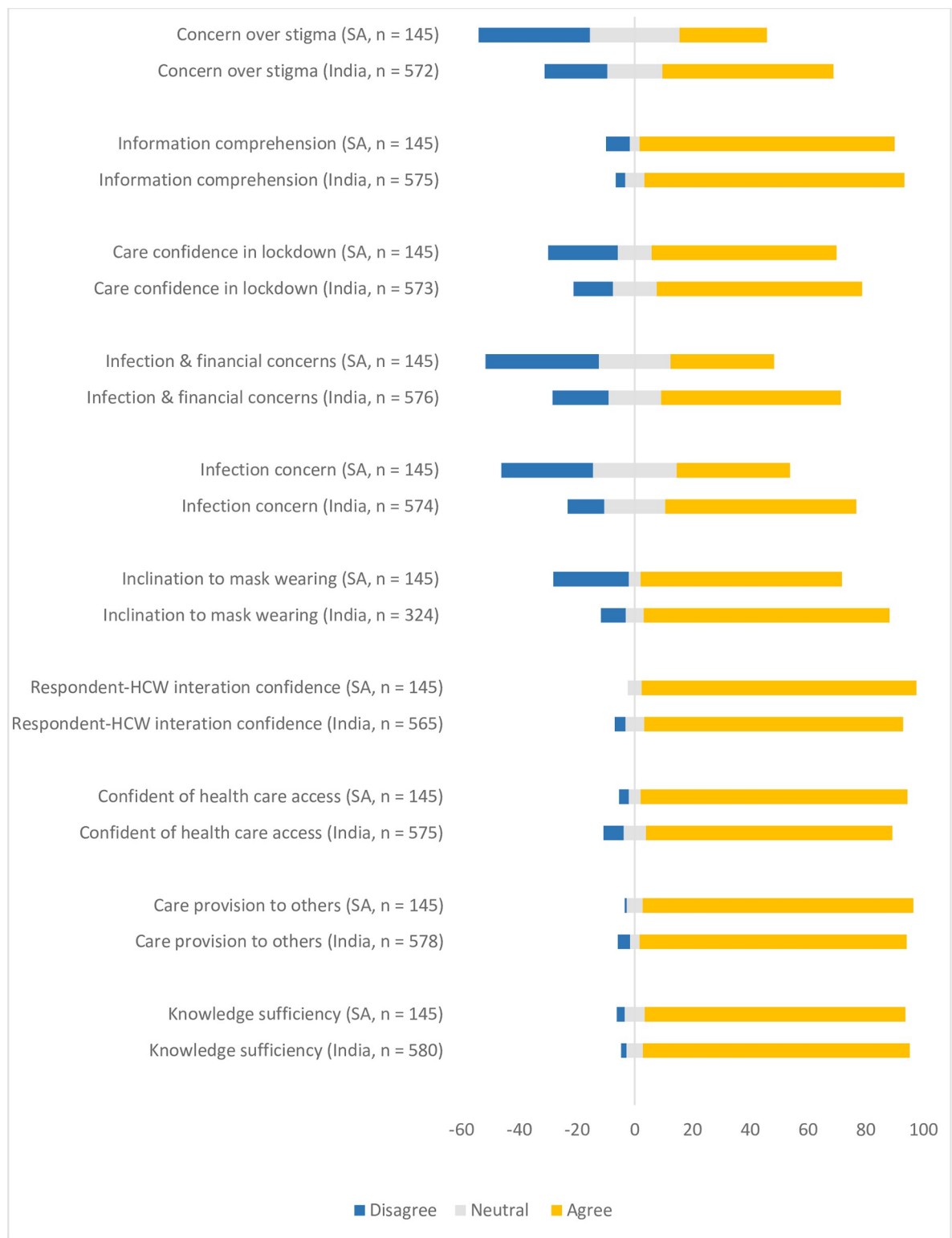

**Fig 2. Respondents' perceptions of self-efficacy in relation to coping with the COVID-19 pandemic in South Africa (SA) and India.** For each aspect, the proportion of participants who disagreed (blue), had neutral views (grey), or agreed (yellow) are presented.

**Table 3. Relationships between selected variables (influence of socio-demographics on infection prevention behaviors).**

| Query | | India | | | | South Africa | | | |
|---|---|---|---|---|---|---|---|---|---|
| | | Yes | No | N/A | *p*-value | Yes | No | N/A | *p*-value |
| **A: Is hand washing frequency affected by water supply?** | Wash more | 476 | 79 | | 0.2168 | 140 | 3 | | 0.7970 |
| | Wash less | 13 | 0 | | | 0 | 0 | | |
| | No change | 15 | 2 | | | 22 | 1 | | |
| | Other | 2 | 0 | | | 1 | 0 | | |
| | Don't know | 5 | 1 | | | 0 | 0 | | |
| | Multiple | 10 | 5 | | | 0 | 0 | | |
| **B: Is water supply affected by employment?** | Student | 124 | 9 | | 0,0008 | 5 | 0 | | 0,4471 |
| | Part time | 27 | 8 | | | 11 | 0 | | |
| | Full time | 217 | 29 | | | 62 | 0 | | |
| | Self-employed | 40 | 7 | | | 47 | 3 | | |
| | Unemployed | 101 | 20 | | | 12 | 1 | | |
| | Retired | 18 | 8 | | | 19 | 0 | | |
| | Other | 8 | 2 | | | 1 | 0 | | |
| | Laid off | 4 | 4 | | | 5 | 0 | | |
| **C: Is avoidance of healthcare facilities because of fear of COVID-19 contraction influenced by age?** | <20 | 11 | 15 | 11 | 0.3217 | 2 | 1 | 1 | 0.6113 |
| | 20–29 | 83 | 147 | 42 | | 4 | 4 | 2 | |
| | 30–39 | 28 | 81 | 9 | | 7 | 9 | 4 | |
| | 40–49 | 24 | 49 | 4 | | 21 | 18 | 12 | |
| | 50–59 | 11 | 31 | 3 | | 17 | 19 | 8 | |
| | 60–69 | 10 | 19 | 1 | | 8 | 14 | 3 | |
| | > = 70 | 2 | 10 | 0 | | 1 | 5 | 1 | |
| **D: Is avoidance of health care facilities because of fear of COVID-19 contraction influenced by employment?** | Student | 37 | 57 | 27 | 0,3143 | 2 | 1 | 1 | 0.1686 |
| | Part time | 12 | 20 | 1 | | 5 | 3 | 3 | |
| | Full time | 57 | 137 | 27 | | 25 | 21 | 11 | |
| | Self-employed | 11 | 28 | 2 | | 10 | 26 | 10 | |
| | Unemployed | 31 | 79 | 5 | | 5 | 6 | 1 | |
| | Retired | 9 | 17 | 0 | | 8 | 7 | 2 | |
| | Other | 5 | 3 | 1 | | 0 | 1 | 0 | |
| | Laid off | 1 | 4 | 3 | | 0 | 2 | 2 | |

pandemic [10]. The aim was to gain some understanding of knowledge and views about the pandemic, particularly when considering the expected roles that the public have in this pandemic regarding social distancing and infection prevention through hand hygiene, mask use and vaccination uptake.

The sampling shows a major skew of respondents towards India. We believe that use of the paper version of the form in India (which could not be replicated for South Africa at the time of data collection) allowed the research team in India to personally engage with potential participants which may have influenced and increased the participant pool in India (Table 1). While the intention was to collect data with increased representation across both countries, this was not readily achieved. The survey invitation was distributed through the network of the researchers who are largely healthcare workers–some of whom were involved in the pandemic response. Majority of the respondents to this survey of patients and the public were resident in the province or state where the research team worked (Table 1); as such, the results discussed here are not necessarily representative of views from residents of the two countries.

At the time of the study, these two countries were at different phases of the pandemic infection curves with no viable vaccines available. Although these data are somewhat dated, these findings add to the body of knowledge on the public's perceptions of the pandemic. They also provide information that can be leveraged for improved infection prevention and behavioural interventions–for this and future infectious disease pandemics. Such knowledge will be helpful in infectious disease pandemic control and mitigation, including in the ongoing COVID-19 pandemic. The insights from this study can assist with measures to address continued vaccine hesitancy and inequity when many countries are dealing with a fourth or subsequent infection wave.

From the onset of the pandemic, efforts have been communicated to inform the public of infection risks and required containment/mitigation measures. The predisposition of an individual to comply with infection prevention measures depends in part on their awareness of infection risks. The need for public engagement and hygiene intervention, behaviour change, and consideration of socio-cultural aspects in public awareness initiatives in India and South Africa has been noted in the literature [15–19].

Participants seemed quite knowledgeable about the COVID-19 pandemic, with most correctly identifying infection sources, risks and symptoms. This could be from the volume of news media dedicated to the pandemic which may also have served to provide education and awareness among the public.

Survey respondents demonstrated awareness of the pandemic, with most identifying the primary routes of transmission, incubation period, symptoms of infection, and recommended measures for infection prevention and management of mild conditions, including the reason for and duration of isolation. Information on the pandemic was generally gained from traditional and social media, family and friends, and government websites. Respondents' reliance on general and social media as sources of pandemic-related information highlights the role played by the media in pandemic containment and mitigation. There was no significant association between positive predisposition to infection prevention measures (such as COVID-19 vaccination) and socio-demographic factors, in contrast with findings in previous studies, perhaps because our sample was biased towards the highly educated [20, 21].

The information landscape has changed extensively in the last three decades, prompting the need to address not only the SARS-CoV-2 pandemic but also its related infodemic [22]. While the main aspects of an infodemic refer to inaccurate and misleading information shared through digital and physical environments during disease outbreaks, disinformation refers to the deliberate spread of false information. In this pandemic, we are increasingly witnessing a growing infodemic driven by misinformation, including a worrying trend in the escalation of disinformation through social and digital media [23–25]. The role of the media, traditional and digital alike, in framing and rapidly disseminating information is evident in this pandemic, particularly when related to influencing behaviours and empowering individuals with the accurate information to make informed decisions regarding IPC [22, 26–28].

Family and friends were noted as sources of SARS-CoV-2 information by respondents in the survey. Word of mouth presented face-to-face or through various communication channels within families and among friends, though not specifically a media source, is an essential source of information. It is also a key route for spreading misinformation, mainly because of the trust between the source and the recipient. Thus, the prominence of influencers (in the community and on digital platforms alike) in disseminating pandemic-related information is highlighted.

The findings of this study underscore the importance of various media as sources of information for informed decision-making among the public. It also draws attention to the relevance of social media, and family and friends, as sources of pandemic-related information for

the public. Given the infodemic that has trailed the SARS-CoV-2 pandemic on all media [28, 29], there is a need for evidence-informed and timely communication in continually addressing pandemic-related misinformation and disinformation. Infodemic management is multifaceted, requiring different disciplines to address it. Beyond communication, factors influencing an individual's behaviours may relate to external pressures, including the economy, education, health literacy, cultural or other beliefs [30], which may be helpful to explore in further studies.

Some respondents in this study considered SARS-CoV-2 to be food- or water-borne. Such beliefs may impact infection prevention measures; while there has been research into transmission by these routes [31, 32], they have not been noted as primary transmission routes for the viral infection. Droplet and airborne transmission have been noted as some primary transmission routes, with the use of face masks a significant intervention in reducing the spread of the infection [33, 34].

Across both countries, some respondents expressed some level of reluctance to mask-wearing, despite their concern about contracting the infection, which may be related to the stigma or discomfort of masks. Stigma, known to influence/compromise infection prevention behaviours [35, 36], needs to be addressed, locally and globally, not only for the current pandemic but also for future ones, to improve adherence to optimised infection prevention practices.

Among other options to reduce infection risk, hand hygiene has been prioritised in public health messages for pandemic mitigation [37]. Access to clean water is critical for hand hygiene and is among the tools to address and mitigate the impact of the pandemic, as highlighted in the literature [37, 38]. There was no relationship between awareness and practice of infection prevention measures (such as hand hygiene)) and socio-demographic factors such as employment status or availability of amenities (such as water supply) (Table 3). While infection prevention measures such as hand hygiene and physical distancing may pose a challenge in some LMIC (India and South Africa are examples), especially in under-resourced sections of rural areas or densely populated urban settings [37, 38], water supply did not affect hand hygiene frequency among our survey respondents.

Isolation and quarantine of infected and exposed individuals are underlying measures for infectious disease control, though this may prove challenging. Responses to SARS-CoV-2-related isolation/quarantine duration reflect respondents' perceptions of SARS-CoV-2 incubation. While there was an initial consensus on a 14-day isolation/quarantine period for infected/affected individuals, there have been shifts and debates on the optimum incubation period of the virus, hence, the duration of isolation and quarantine measures [39]. Respondents' responses reflected this, more so in India, where discussions about extended isolation periods have been reported [40].

Lockdown measures instituted in various parts of the world following the spread of SARS-CoV-2 served as another infectious disease mitigation strategy. With the rise of infection transmission and the attendant lockdown measures, it was expected that individuals would have avoided visiting healthcare facilities. Some participants in this study reported avoiding healthcare facilities because of a fear of contracting the COVID-19 virus; with current data, it couldn't be concluded that this was influenced by employment status although a case could be made for South Africa as it had a more significant relationship than India despite there being less South African participants than Indian participants. Employed participants may be more likely motivated to maintain good health or hesitant to confirm illness, for fear of losing money or work, resulting in fewer visits to healthcare facilities, than those unemployed. While lockdown measures can reduce patient presentation to healthcare facilities [41, 42], such a decline in presentation may also be associated with later presentations with more severe

consequences. Initiatives are required to address gaps in patient care necessitated by public health promotion strategies such as lockdowns in this and future pandemics.

Across the two countries, attitudes to COVID-19 vaccination were positive (Table 2). However, the country analysis showed this was driven by higher vaccine acceptance in India, with respondents in South Africa more cautious regarding COVID-19 vaccination. The significant difference in vaccine acceptance / hesitancy between those who were vaccinated and those who were not vaccinated against the influenza virus (p = 0.003) has been reported in another study where knowledge of COVID-19 vaccination was noted to be associated with past experience of vaccine uptake [43]. Reasons cited for hesitancy or a negative attitude to SARS-CoV-2 vaccination were related to mistrust in the vaccine development process and the futility of vaccines for what respondents considered a self-limiting flu-like illness.

This survey was, however, conducted before SARS-CoV-2 vaccines were available; perceptions and attitudes may have changed in the time since the survey was conducted. Hesitancy towards the SARS-CoV-2 vaccine had been noted earlier in the pandemic, fuelled by the circulation of SARS-CoV-2-related conspiracies [44, 45] and associated with various socio-demographic and other factors [46–48]. The notion that the pandemic has been grossly exaggerated and reported, with unnecessary financial and other stresses on populations, was expressed by some participants who provided additional free text information across both countries. As the pandemic evolves, research to better understand infection and vaccine-related concerns among the general population is needed to support targeted and contextually appropriate strategies promoting vaccine uptake and optimised infection prevention behaviours.

Among individuals with opposing opinions about vaccination, application of social science methods to study underlying reasons and contexts for their views, along with highlighting the individual rather than the collective advantages of vaccination, may provide helpful and relatable insight [45, 49]. This could be particularly important when considered in light of recent research and noted factors that may influence vaccine perception and uptake [45–50]. More recent research has provided insight into dealing with vaccine-hesitancy as well as the challenges associated with anti-vaxxers [45, 48]. Public health campaigns and vaccination promotions should therefore understand and leverage social listening techniques to comprehend public perceptions concerning communication gaps. A similar method of social listening should be developed for community and traditional settings to understand why various beliefs and behaviours related to COVID-19 emerged.

## Strengths and limitations

Our study provides unique insights into the public's attitudes and practices across two LMIC during the early stages of this pandemic. The findings are subject to some limitations, which should be considered in the interpretation.

First, being a cross-sectional study, it cannot be used to determine temporal relationships. In addition, the relevance of the findings may change over time and with interventions, especially as subsequent waves of COVID-19 have been reported. Second, the online distribution of the survey and the limited paper version may have limited its reach, particularly under-representing individuals from diverse socio-economic levels. In addition, data collection across both sites did not rely on the same methods, given the COVID-19 restrictions at the time of data collection, which likely influenced the sample sizes across the sites. Survey respondents are therefore not representative of the public in either of the two countries, limiting the generalizability of findings.

We utilized COVID-19 vaccine uptake views as a proxy for pandemic awareness. When looking at perceptions and awareness, some questions in the survey were related to this. There

may not have been a simply way to amalgamate them to categorise participants in levels of either. Our assumption was that if people were willing to take the vaccine, then they were aware of why the vaccine was necessary as they perceived it as something positive whereas if someone was unwilling to take the vaccine, they were assumed not to be aware of its importance and therefore have a low level of COVID-19 perception and awareness. Of course, with this assumption, certain groups of people will be misclassified; an example could be those who are COVID-19 aware but refuse the vaccine for religious reasons. On the other hand, there may also be those that have no knowledge of COVID-19 but are willing to get a vaccine only because they understand the need for a vaccine (for work or other reasons) and not necessarily because they understand the dangers of COVID-19. There isn't always a fine line between perception and awareness and so the binary question was used to encapsulate the topics and produce a workable statistic.

Nevertheless, this paper fills a gap in the knowledge, awareness and attitudes of a section of the public in India and South Africa towards IPC practices in the context of COVID-19 within the first year of the pandemic. It will be beneficial for charting public understanding and perception of the COVID-19 pandemic and provides informative data that can be employed for public engagement in other infectious disease control and mitigation efforts, across both sites and similar contexts. The skew of respondent sample towards the highly educated and those in contact with healthcare services or professionals (given that recruitment was through the network of the research team) means that while findings are biased, they are likely to provide a reasonable picture of the 'best case' scenario as these respondents will likely be more knowledgeable and better informed than the average general respondent.

While this research presents the data for each country separately, it is not its intention to make any statistical comparisons between participants in the two countries. Despite that, the individual test on how employment affects water supply provided some insight on differences between participants in the two countries. Thus, the need for pandemic mitigation efforts to consider differences in context and subjects for the delivery of context-specific and appropriate interventions is highlighted.

## Recommendations

This study presents socio-economic and demographic data, which may influence public awareness and behaviour and further be explored in pandemic mitigation initiatives among the public in both countries. Survey respondents correctly identified public health promotion measures for SARS-CoV-2. Reported disinclination to mask-wearing and reported hesitancy for the uptake of SARS-CoV-2 vaccination highlight gaps that can be addressed for improved pandemic mitigation efforts. While the data is not representative of participants across both countries, further research to explore the outlook towards mask use and vaccination across both countries can provide more insight on factors influencing infection prevention and vaccine hesitancy. Vaccination campaigns should consider robust public engagement and more targeted communication strategies using tactics like social listening, with multimodal, participatory online and offline initiatives to engage academics, health care providers, and the society in curbing the pandemic.

## Supporting information

**S1 Checklist. Reporting checklist for cross-sectional study.**
(DOCX)

**S1 Data.**
(XLSX)

**S2 Data. Chi-squared tests.**
(XLSX)

**S3 Data. Regression tests.**
(XLSX)

## Acknowledgments

The authors express appreciation to all survey participants and members of the public who participated in the review of or provided feedback on the survey tool, and to Ms Jean Fourie for review and editing of the manuscript.

## Author Contributions

**Conceptualization:** Oluchi Mbamalu, Sanjeev Singh, Esmita Charani, Marc Mendelson.

**Data curation:** Oluchi Mbamalu, Surya Surendran, Vrinda Nampoothiri, Fabia Edathadathil, Nina Zhu, Esmita Charani.

**Formal analysis:** Oluchi Mbamalu, Fabia Edathadathil, Ebrahim Steenkamp, Esmita Charani.

**Funding acquisition:** Oluchi Mbamalu, Alison Holmes, Marc Mendelson.

**Investigation:** Oluchi Mbamalu, Surya Surendran, Vrinda Nampoothiri, Candice Bonaconsa, Fabia Edathadathil, Nina Zhu, Vanessa Carter, Helen Lambert, Raheelah Ahmad, Adrian Brink, Alison Holmes, Sanjeev Singh, Esmita Charani, Marc Mendelson.

**Methodology:** Oluchi Mbamalu, Surya Surendran, Vrinda Nampoothiri, Candice Bonaconsa, Nina Zhu, Vanessa Carter, Helen Lambert, Carolyn Tarrant, Raheelah Ahmad, Adrian Brink, Sanjeev Singh, Esmita Charani, Marc Mendelson.

**Project administration:** Oluchi Mbamalu, Surya Surendran, Candice Bonaconsa, Alison Holmes, Sanjeev Singh, Esmita Charani, Marc Mendelson.

**Resources:** Alison Holmes, Sanjeev Singh, Marc Mendelson.

**Supervision:** Oluchi Mbamalu, Sanjeev Singh, Esmita Charani, Marc Mendelson.

**Validation:** Oluchi Mbamalu, Surya Surendran.

**Visualization:** Oluchi Mbamalu, Fabia Edathadathil, Ebrahim Steenkamp, Esmita Charani.

**Writing – original draft:** Oluchi Mbamalu, Surya Surendran.

**Writing – review & editing:** Oluchi Mbamalu, Surya Surendran, Vrinda Nampoothiri, Candice Bonaconsa, Fabia Edathadathil, Nina Zhu, Vanessa Carter, Helen Lambert, Carolyn Tarrant, Raheelah Ahmad, Adrian Brink, Ebrahim Steenkamp, Alison Holmes, Sanjeev Singh, Esmita Charani, Marc Mendelson.

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
