## [Decision Letter · Decision Letter 0]

2 Oct 2022

PGPH-D-22-01370

A survey of patient and public perceptions and awareness of SARS-CoV-2-related risks among participants in India and South Africa

Dear Dr. Mbamalu,

Thank you for submitting your manuscript to PLOS Global Public Health. After careful consideration, we feel that it has merit but does not fully meet PLOS Global Public Health’s publication criteria as it currently stands. Therefore, we invite you to submit a revised version of the manuscript that addresses the points raised during the review process.

We have four excellent reviews. Many of them brought up similar points in regards to sampling. If there are any contradictions across reviewers, please just explain why you have made a certain choice in editing the manuscript.

We look forward to receiving your revised manuscript.

Kind regards,

Abram L. Wagner, PhD, MPH

Academic Editor

Journal Requirements:

1. We have amended your Competing Interest statement to comply with journal style. We kindly ask that you double check the statement and let us know if anything is incorrect. 

2. Please provide a/amend your detailed Financial Disclosure statement. This is published with the article. It must therefore be completed in full sentences and contain the exact wording you wish to be published.

3.Please provide separate figure files in .tif or .eps format.

4. We have noticed that you have uploaded Supporting Information files, but you have not included a list of legends. Please add a full list of legends for your Supporting Information files after the references list. 

5. In the online submission form, you indicated that "Data is available on a secure online portal, and can be accessed on request or included as supplementary information, if preferred." All PLOS journals now require all data underlying the findings described in their manuscript to be freely available to other researchers, either 1. In a public repository, 2. Within the manuscript itself, or 3. Uploaded as supplementary information.

Additional Editor Comments (if provided):

Reviewers' comments:

Reviewer's Responses to Questions

**Comments to the Author**

1. Does this manuscript meet PLOS Global Public Health’s publication criteria? Is the manuscript technically sound, and do the data support the conclusions? The manuscript must describe methodologically and ethically rigorous research with conclusions that are appropriately drawn based on the data presented.

Reviewer #1: No

Reviewer #2: Partly

Reviewer #3: Partly

Reviewer #4: Partly

2. Has the statistical analysis been performed appropriately and rigorously?

Reviewer #1: No

Reviewer #2: I don't know

Reviewer #3: Yes

Reviewer #4: I don't know

3. Have the authors made all data underlying the findings in their manuscript fully available (please refer to the Data Availability Statement at the start of the manuscript PDF file)?

Reviewer #1: No

Reviewer #2: Yes

Reviewer #3: Yes

Reviewer #4: Yes

4. Is the manuscript presented in an intelligible fashion and written in standard English?

Reviewer #1: Yes

Reviewer #2: Yes

Reviewer #3: Yes

Reviewer #4: Yes

5. Review Comments to the Author

Reviewer #1: Thank you for the chance to review the paper.

The authors present original research collected from a part of India and in South Africa in late 2022 to assess public perception attitudes and practices related to COVID-19. survey data is used to provide social demographic descriptive details of the population from India and South Africa and chi-squared analysis is performed to identify variables linked with sources of knowledge behaviours and select COVID-19 safe practices.

While the paper presents findings from before the currently COVID-19 waves from 2021 and beyond the paper's insights provide a valuable addition to the existing literature on public perception of COVID-19 and how it shifted over time.

The paper however requires revisions that are substantive in nature across the introduction methodology for the formatting of results and discussion before it can be considered for potential publication.

1. The introduction does not provide an adequate summary of the literature or provide an adequate rationale for why this study has been conducted. India and SA have had very different COVID experiences I could not understand why these countries were selected by the authors for the paper.

2. The methodology is a major concern of the paper as adequate details of how the sample was estimated, how the sampling framework was drawn and how people were identified to participate in the study are not provided.

3. The current sampling shows a major skew of respondents towards India needs to be discussed as it has affected the results, moreover, it seems data was only collected from a specific region or state of India and not whole of India. The authors are advised to correct details wherever applicable to specify that the study is only from a select population

4. While Chi square analysis provides insights I wonder why the authors did not consider other methodologies such as regression analysis to estimate what the determinants of knowledge might have been as well as determinants of COVID safe behaviours practice.

5. I places the presentation of results detract from the value of the paper specifically in the sections linked to employment, hand washing and sources of water. The tables here need to be remade and I suggest only using one table for all of the information as well as providing better insights as to why these associations are important.

6. Hand washing is universally different depending on not only social economic status and access to water but also perceptions related to hand washing, the authors could bring in how hand washing in COVID is different from non-COVID related situations.

7. I would suggest the authors also separate out specific factors related to India and South Africa and present them differently as there are various cultural social economic and other factors that influences study outcomes

Reviewer #2: For the paper, you need to be concerned about the methods section e.g. study settings and study procedure. Data collection i.e who has collected data and their training should be mentioned. Please find my minor comments in the attached file.

Reviewer #3: Summary of the Research:

This cross-sectional study (online and offline survey) has tried to explore the perception and awareness of SARS CoV-2 related risks among patients and general public (>18 yrs of age) in South Africa and India using a self –administered questionnaire. A total of 844 respondents (India-660 and South Africa-184) responded to the survey. This survey was conducted during September 2020- December 2020 (during 1st wave of covid in India and 2nd wave in South Africa). The study found that the respondents were aware about the public health promotion measures of SARS CoV-2 and vaccine hesitancy was reported to be more among people in South Africa than in India.

General remarks

The research team has made tremendous efforts in collecting data through offline and online mode in both the countries during the peak pandemic period. However since the data was collected during the initial phase of the pandemic and both the countries has faced many waves after that, the relevance and importance of the data (knowledge obtained from the study) in the current context and its contribution in initiating any public health measures is the matter of concern.

Study setting & participants:

Describe the study setting in detail

*Whether online survey covered participants from all over India and South Africa or covered only a part of these countries?

*Where was the offline survey conducted? (exact location, rural/urban setting)

*Who were invited to participate in the online survey?

* Since the study made use of a self-administered questionnaire to assess the knowledge, clarify whether illiterates were included/excluded from the study

* Since the questionnaire was translated only in Malayalam apart from English in India, clarify whether the survey included only the participants who can read and write either in English/Malayalam. Whether people who knows other Indian languages were included or not.

Sample size

No sample size calculation- Provide Justification for the sample size included in the study

Ethics statement

Clarify whether any signature/thumb impression was obtained from the participants apart from checking the relevant boxes for obtaining consent for offline participants? If, so justify why it was done so and whether permission was obtained from Ethics Committee to obtain informed consent in that way

Statistical analysis

Mention if any statistical software was used for data analysis?

Results

Table 1 -Clarify what is included in others in education (is it illiterates?)

Table 3- Table title is incomplete

Figures are not explained in text part

Discussion

Discussion part concentrates more about the source of information, rather than the knowledge and perception about SARS CoV-2 risk, which is the main objective

Conclusions

In line no.45, the authors have claimed that this study presents socio-economic and demographic data which may influence public awareness and behaviour. Since the authors has collected data mostly through online mode, there is a high possibility that only well educated people would have responded to the survey and this might have led to under representation of the individuals from diverse socio-economic class (which is acknowledged by the authors themselves in the limitation section). Hence such a conclusion cannot be derived from the present study

Reviewer #4: Review of “A survey of patient and public perceptions and awareness of SARS-CoV-2-related risks among participants in India and South Africa’’ submitted to PLOS Global Public Health

The paper covers a topical issue, especially as the Covid-19 pandemic is still evolving in different ways across different countries, with unequal effects on low- and upper-income countries.

However, I do not think it sufficiently meets the standard required for publication at this point.

I will therefore offer below, suggestions for improving the quality of the manuscript, which I think would make it suitable for publication.

The authors should be commended for their efforts in carrying out a multi-country study at the peak of the Covid pandemic.

Overall:

The cross-sectional study would benefit from the use of a theoretical framework to help establish validity. With no clear evidence of randomization in its sampling, the use of a theoretical framework will strengthen the article. Also, a more detailed explanation of its statistical analysis is required. Furthermore, attention should be paid to grammatical constructions, as there are sentences that do not convey clear or appropriate meanings, which negatively impacts on the article.

I have only restricted myself to commenting on the sections where I think corrections should be made. In sections where I have not commented, the authors conformed with the minimum standards required, in my opinion.

Abstract

The abstract should state the conclusions regarding the main outcome measures.

Study design

-The statement ‘’The study development followed the STROBE cross-sectional reporting guidelines ‘’ is confusing as STROBE appears to be a reporting guideline and not for study development. This should be clarified by the authors or changed

-The domains identified in the survey instrument ‘’perceived risk, barriers and self-efficacy’’ appeared to be that of the health belief model of health promotion. There is need to expatiate on this theoretical framework used in developing the survey. Authors should thus dedicate a section to explain this model, or any other model used and how it was adapted to the study.

Study setting and participant

The authors stated that all invited individuals received participant information leaflets……..

How were these participants invited? For the paper-based survey especially, authors need to state how these individuals were invited or recruited to participate in the study

Data collection

It is good to say more about the [hospital] study site. How big is the hospital? is it a tertiary hospital? If it is a multi-unit hospital from which of the units were participants selected? For example, if the hospital in this case was a Covid isolation center, it’s good to know as this may influence participants’ responses etc. In addition, was patient selection randomized? If not, what efforts were made to ensure that a single individual did not fill more than one survey instrument?

Statistical analysis

Which statistical software was used to analyze the data? What version?

Results

-The authors stated that ‘’the response rate for the paper version of the survey could not be estimated, as respondents returned a higher number of the completed survey forms than the initial number disseminated, indicating the forms had been copied and shared more widely.’’

It is good to state here what the initial number of survey forms distributed was.

-The statement ‘’There was no statistical significance between hand washing and water supply (Table 3)……’’ should be rephrased as it lacks meaning in its current form. I assume the authors meant there was no association.

-In Table 3 C, the authors alluded to age influencing avoidance of healthcare facilities by respondents

In what direction is this association? Is it more likely in young people or older people?

Discussions

The sentence that starts with ‘’Also, how a better understanding of this information can be leveraged……..’’ should be rephrased to give it meaning. In its current form, it is neither a direct continuation of the last sentence nor meaningful on its own

Limitations

It should be stated at the outset that a cross sectional study’s main limitation is that it cannot determine temporal relationships.

The authors did well to have stated the other limitations, including the non-generalizability of findings because most respondents are young, educated people who are likely to be more informed about the SARS CoV 2, hence are not representative of the public

6. PLOS authors have the option to publish the peer review history of their article (what does this mean?). If published, this will include your full peer review and any attached files.

**Do you want your identity to be public for this peer review?** For information about this choice, including consent withdrawal, please see our Privacy Policy.

Reviewer #1: **Yes: **Dr Danish Ahmad

Reviewer #2: No

Reviewer #3: No

Reviewer #4: No

---

## [Editor Report · Decision Letter 1]

21 Dec 2022

PGPH-D-22-01370R1

A survey of patient and public perceptions and awareness of SARS-CoV-2-related risks among participants in India and South Africa

Dear Dr. Mbamalu,

Thank you for submitting your manuscript to PLOS Global Public Health. After careful consideration, we feel that it has merit but does not fully meet PLOS Global Public Health’s publication criteria as it currently stands. Therefore, we invite you to submit a revised version of the manuscript that addresses the points raised during the review process.

I just had minor comments about formatting. I can process a decision quickly when you respond to these.

We look forward to receiving your revised manuscript.

Kind regards,

Abram L. Wagner, PhD, MPH

Academic Editor

Journal Requirements:

2. We do not publish any copyright or trademark symbols that usually accompany proprietary names, eg  ©, ®, ™  (e.g. next to drug or reagent names). Please remove all instances of trademark/copyright symbols throughout the text, including ® on pages 13.

Additional Editor Comments (if provided):

Table 3 - could you move the answer choices to the left of the India column?

Figure 2 - why is this centered around the middle of neutral? Additionally, my recommendation is to condense some of the bar charts and increase font size. The caption/legend does not need to be in the TIF image itself.

Can you mention the word "convenience sampling" in your methods where you talk about your sample design?
---

## [Decision Letter · Decision Letter 2]

27 Jan 2023

PGPH-D-22-01370R2

A survey of patient and public perceptions and awareness of SARS-CoV-2-related risks among participants in India and South Africa

Dear Dr. Mbamalu,

Thank you for submitting your manuscript to PLOS Global Public Health. After careful consideration, we feel that it has merit but does not fully meet PLOS Global Public Health’s publication criteria as it currently stands. Therefore, we invite you to submit a revised version of the manuscript that addresses the points raised during the review process.

The reviewer comments are generally favorable. Please go through the manuscript and be a bit more detailed in describing the state/region in India you select from instead of referring to India more generally. This is a minor change and I'll be able to process your manuscript quickly when you send it back.

We look forward to receiving your revised manuscript.

Kind regards,

Abram L. Wagner, PhD, MPH

Academic Editor

Journal Requirements:

Additional Editor Comments (if provided):

Reviewers' comments:

Reviewer's Responses to Questions

**Comments to the Author**

1. If the authors have adequately addressed your comments raised in a previous round of review and you feel that this manuscript is now acceptable for publication, you may indicate that here to bypass the “Comments to the Author” section, enter your conflict of interest statement in the “Confidential to Editor” section, and submit your "Accept" recommendation.

Reviewer #1: (No Response)

Reviewer #3: All comments have been addressed

Reviewer #4: (No Response)

2. Does this manuscript meet PLOS Global Public Health’s publication criteria? Is the manuscript technically sound, and do the data support the conclusions? The manuscript must describe methodologically and ethically rigorous research with conclusions that are appropriately drawn based on the data presented.

Reviewer #1: Yes

Reviewer #3: Yes

Reviewer #4: Partly

3. Has the statistical analysis been performed appropriately and rigorously?

Reviewer #1: Yes

Reviewer #3: Yes

Reviewer #4: I don't know

4. Have the authors made all data underlying the findings in their manuscript fully available (please refer to the Data Availability Statement at the start of the manuscript PDF file)?

Reviewer #1: Yes

Reviewer #3: Yes

Reviewer #4: Yes

5. Is the manuscript presented in an intelligible fashion and written in standard English?

Reviewer #1: Yes

Reviewer #3: Yes

Reviewer #4: Yes

6. Review Comments to the Author

Reviewer #1: The revised paper is much improved. The authors have addressed my comments but there is one important comment that they explain in their response has not been made in the paper itself.

The major revision is for the authors to add their (adapted)response to my comment belowin the paper itself

I encourage the authors to add pertinent details to the paper itself for my comment related to the following 'The current sampling shows a major skew of respondents towards

India needs to be discussed as it has affected the results, moreover, it seems data was only collected from a specific region

or state of India and not whole of India. The authors are advised to correct details wherever applicable to specify that the study is only from a select population'

Reviewer #3: (No Response)

Reviewer #4: All my comments in the last review have been properly addressed.

However, The authors should please clarify:

In line 56/57, you mentioned that your main outcome measures were awareness of SARS-CoV-2-related risks and perception of infection risks.

However, between lines 261-265, in your logistic regression analysis, your response variable which should be your outcome or dependent variable was Covid-19 vaccination (‘if people were willing to take the vaccine'=1, if not=0)

Can you please explain this discrepancy? Was Covid-19 vaccine uptake used as proxy for awareness level? If not, how was awareness measured or categorized for logistic regression? e.g high level or low level, aware or unaware?

7. PLOS authors have the option to publish the peer review history of their article (what does this mean?). If published, this will include your full peer review and any attached files.

**Do you want your identity to be public for this peer review?** For information about this choice, including consent withdrawal, please see our Privacy Policy.

Reviewer #1: **Yes: **DR Danish Ahmad MBBS(Delhi Uni),MSc(Oxford Uni)MNAMS(India),PhD(Uni>Canberra0

Reviewer #3: No

Reviewer #4: No

---

## [Editor Report · Decision Letter 3]

10 Feb 2023

A survey of patient and public perceptions and awareness of SARS-CoV-2-related risks among participants in India and South Africa

PGPH-D-22-01370R3

Dear Dr Mbamalu,

We are pleased to inform you that your manuscript 'A survey of patient and public perceptions and awareness of SARS-CoV-2-related risks among participants in India and South Africa' has been provisionally accepted for publication in PLOS Global Public Health.

Best regards,

Abram L. Wagner, PhD, MPH

Academic Editor
